# Influence of Seasonality and Pollution on the Presence of Antibiotic Resistance Genes and Potentially Pathogenic Bacteria in a Tropical Urban River

**DOI:** 10.3390/antibiotics14080798

**Published:** 2025-08-05

**Authors:** Kenia Barrantes-Jiménez, Bradd Mendoza-Guido, Eric Morales-Mora, Luis Rivera-Montero, José Montiel-Mora, Luz Chacón-Jiménez, Keilor Rojas-Jiménez, María Arias-Andrés

**Affiliations:** 1Doctorado en Ciencias Naturales para el Desarrollo (DOCINADE), Instituto Tecnológico de Costa Rica, Universidad Nacional and Universidad Estatal a Distancia, San José P.O. Box 11501-2060, Costa Rica; 2Health Research Institute, University of Costa Rica, San José P.O. Box 11501-2060, Costa Rica; bradd.mendoza@ucr.ac.cr (B.M.-G.); luis.riveramontero@ucr.ac.cr (L.R.-M.); jose.montielmora@ucr.ac.cr (J.M.-M.); luz.chacon@ucr.ac.cr (L.C.-J.); 3School of Health Technologies, University of Costa Rica, San José P.O. Box 11501-2060, Costa Rica; eric.morales@ucr.ac.cr; 4Biology School, University of Costa Rica, San José P.O. Box 11501-2060, Costa Rica; keilor.rojas@ucr.ac.cr; 5Instituto Regional de Estudios en Sustancias Tóxicas (IRET), Universidad Nacional, Campus Omar Dengo, Heredia P.O. Box 86-3000, Costa Rica; maria.arias.andres@una.ac.cr

**Keywords:** metagenomics, plasmidome, mobile genetic elements, environmental microbiology, riverine ecosystems, qPCR, public health

## Abstract

Background/Objectives: This study examines how seasonality, pollution, and sample type (water and sediment) influence the presence and distribution of antibiotic resistance genes (ARGs), with a focus on antibiotic resistance genes (ARGs) located on plasmids (the complete set of plasmid-derived sequences, including ARGs) in a tropical urban river. Methods: Samples were collected from three sites along a pollution gradient in the Virilla River, Costa Rica, during three seasonal campaigns (wet 2021, dry 2022, and wet 2022). ARGs in water and sediment were quantified by qPCR, and metagenomic sequencing was applied to analyze chromosomal and plasmid-associated resistance profiles in sediments. Tobit and linear regression models, along with multivariate ordination, were used to assess spatial and seasonal trends. Results: During the wet season of 2021, the abundance of antibiotic resistance genes (ARGs) such as *sul-1*, *intI-1*, and *tetA* in water samples decreased significantly, likely due to dilution, while *intI-1* and *tetQ* increased in sediments, suggesting particle-bound accumulation. In the wet season 2022, *intI-1* remained low in water, *qnrS* increased, and sediments showed significant increases in *tetQ*, *tetA*, and *qnrS*, along with decreases in *sul-1* and *sul-2*. Metagenomic analysis revealed spatial differences in plasmid-associated ARGs, with the highest abundance at the most polluted site (Site 3). Bacterial taxa also showed spatial differences, with greater plasmidome diversity and a higher representation of potential pathogens in the most contaminated site. Conclusions: Seasonality and pollution gradients jointly shape ARG dynamics in this tropical river. Plasmid-mediated resistance responds rapidly to environmental change and is enriched at polluted sites, while sediments serve as long-term reservoirs. These findings support the use of plasmid-based monitoring for antimicrobial resistance surveillance in aquatic systems.

## 1. Introduction

Urban aquatic environments in tropical regions are subject to growing anthropogenic pressures, including untreated wastewater discharges, agricultural runoff, and urban development. These stressors alter microbial communities and promote the proliferation of antibiotic resistance genes (ARGs) and potentially pathogenic bacteria [1,2]. Within these communities, mobile genetic elements (MGEs), particularly plasmids, play a central role by mediating horizontal gene transfer (HGT) and facilitating the spread of ARGs across taxonomically diverse bacterial populations [3,4]. Plasmids confer traits that support bacterial persistence under selective pressures, such as chemical pollutants [3,5].

In tropical watersheds, the resistome—the collection of ARGs within a microbial community—can reflect varying levels of anthropogenic pollution. For instance, a study conducted in a rural area of Costa Rica found that ARGs detected in wildlife samples increased with the level of human activity along the river [6]. In more urbanized settings, the Virilla River watershed showed higher abundance and diversity of ARGs and virulence genes in *Escherichia coli*, particularly during the wet season, associated with urban and grazing land uses [1]. Similarly, isolates from polluted surface waters in this watershed harbored more plasmid-borne ARGs and multidrug resistance profiles, especially downstream of wastewater discharges [2]. A previous study also demonstrated that urban pollution reshaped plasmid community structure independently of chromosomal variation, alongside increased ARG diversity [7].

In addition to pollution, environmental factors as climate are key to understanding the spread of ARGs. In tropical systems, seasonal hydrological variation is a major driver of ARG dynamics [8,9]. During the wet season, increased runoff and sediment resuspension contribute to the mobilization of bacteria, mobile genetic elements (MGEs), and contaminants from urban surfaces and soils into river systems, facilitating bacterial aggregation and horizontal gene transfer (HGT) [10,11]. In contrast, the dry season typically features low water flow and limited dilution, favoring the accumulation of pollutants and selection for resistant microorganisms [1,12].

Sediments play a central role in aquatic systems, acting as long-term reservoirs of contaminants, ARGs, and microbial communities. Due to their capacity to retain pollutants, sediments serve as hotspots for gene exchange and persistence of resistance elements, even when contamination levels in the water column fluctuate. Hydrological disturbances, such as floods or storm-driven flows, can resuspend sediment-bound ARGs, returning them to the water column and increasing the risk of dissemination [7,13]. Understanding these sediment-related processes is essential for assessing the long-term ecological and public health implications of ARG contamination.

Despite their importance, plasmidomes in environmental resistome studies remain understudied, especially in tropical conditions. Distinguishing plasmid-derived from chromosomal sequences within metagenomic datasets allows for a more accurate assessment of ARG distribution and improves our understanding of their ecological dynamics. This approach is supported by recent advances in bioinformatics and machine learning, which enable the improved identification of plasmids in complex environmental samples, revealing their potential as indicators of ecosystem disturbance and reservoirs of antimicrobial resistance [7,14,15].

This study aims to characterize the spatial and seasonal patterns of bacterial communities and antibiotic resistance genes (ARGs) in a tropical urban river in Costa Rica. Metagenomic analyses were conducted on sediment samples, and quantitative PCR (qPCR) was applied to both water and sediment samples, across three sampling campaigns representing distinct hydrological conditions. A main objective was to distinguish ARGs located on plasmids from those on chromosomes and to evaluate how urban pollution and seasonal variability influence their distribution and diversity. Special emphasis was placed on the role of sediments as environmental reservoirs and potential sources of ARG mobilization.

In this context, the *plasmidome* is defined as the complete set of plasmid-derived sequences identified via metagenomic analysis, including both coding and non-coding elements, including ARGs and taxonomically assigned sequences. This is distinct from plasmid-associated ARGs, which specifically refer to resistance genes located on plasmid contigs. By integrating qPCR and metagenomic data across environmental matrices and time points, this study provides a comprehensive assessment of microbial community dynamics and the environmental dissemination of antibiotic resistance in a highly impacted tropical watershed.

## 2. Results

### 2.1. Physicochemical and Microbiological Parameters

A pollution gradient in the Virilla River watershed was determined based on microbiological and physicochemical parameters (Table 1). Significant differences in these parameters were observed among sampling sites only during the 2022 campaigns (Sampling 2 and 3), both in the dry and the wet seasons. In contrast, no significant spatial differences were detected during the first campaign (Sampling 1, wet season 2021) for any of the evaluated parameters (Kruskal–Wallis H test, *p* > 0.05). Overall, during 2022, fecal contamination indicators and other physical parameters were notably higher in Site 2, but even more so at Site 3, compared to Site 1.

During the dry season of 2022 (Sampling 2), significant differences were found across sites for all parameters analyzed. Site 3 showed higher concentrations of fecal coliforms (H = 7.3, *p* = 0.02) and *E. coli* (H = 6.9, *p* = 0.03) compared to Site 1. It also showed greater turbidity (H = 7.2, *p* = 0.02), higher temperature (H = 8.0, *p* = 0.02), and lower dissolved oxygen levels (H = 8.0, *p* = 0.01) than the other sites. These spatial trends remained consistent during wet season 2022. In this campaign, Site 2 showed elevated levels of fecal contamination: fecal coliforms (H test: statistic = 8.0, *p* = 0.01), *E. coli* (H test: statistic = 8.0, *p* = 0.01), and *E. faecalis* (H = 8.0, *p* = 0.01) compared to Site 1. Additionally, Site 3 exhibited significantly higher turbidity (H test: statistic = 7.2, *p* = 0.02) and temperature (H test: statistic = 8.0, *p* = 0.01) than Site 1.

Site 1 consistently maintained better water quality conditions throughout the study period. However, according to NSF-WQI classifications, all sites remained below the “good” quality threshold (70.1–90). Site 1 reached a maximum score of 66 during the dry season of 2022, falling into the “fair” category (50.1–70), while Site 3 consistently showed the lowest scores, for example, 35 during the same campaign, classified as “poor” (25.1–50).

These findings suggest that seasonal variability and localized pollution sources contribute to reduced water quality, particularly at Site 3. Located at a lower elevation along the Virilla River, Site 3 has been previously identified as the most polluted area, with BMWP-CR index scores ranging from 17 to 26 (“very polluted”) and a Dutch Water Quality Index score of 11 (“severe”) [16]. Site 3 also has the lowest NSFWQI values among sampling sites in this study (Table 1).

Overall, these results confirm a gradient of declining water quality within the system, whose intensity varies seasonally and is likely driven by site-specific pollution sources.

### 2.2. Seasonality Influences ARGs Distribution Along the Virilla River Watershed

Seasonal variation in ARG abundance was assessed using Tobit and linear regression models. Tobit models were applied to account for censored data (i.e., gene abundances below detection limits). The 2022 dry season was set as the reference category. The gene *bla_TEM_* was excluded from the regression analyses due to a high proportion of non-detectable values (>80%), while *bla_OXA_* and *bla_CTX-M_* were not detected in any sample. Relative abundances are presented in Figure 1 as geometric means by sampling campaign and site, illustrating seasonal and spatial variations across matrices.

Several ARGs exhibited strong seasonal variation in water samples, as indicated by a high Bayes factor (BF). During the 2021 wet season, *sul-1* (BF = 20.51), *intl-1* (BF = 34.06), and *tetA* (BF = 7.74) were detected at significantly lower concentrations, consistent with visual trends in Figure 1, particularly at Sites 1 and 3. For instance, *intl-1* and *sul-1* were less abundant at these sites in Sampling 1 than in Sampling 2. Although *ermB* showed a moderately high BF (8.11), its confidence interval included zero, and the *p*-value was not significant (Appendix A).

In the 2022 wet season, *intl-1* levels in water samples remained comparatively low (BF = 34.06), whereas *qnrS* was detected at higher levels (BF = 1370). As shown in Figure 1, *intl-1* abundance was lower at Sites 2 and 3 compared to the dry season. No significant seasonal variation was found for *sul-1*, *tetA*, *tetQ*, *sul-2*, or *dfrA12* in Sampling 3. Although *ermB* appeared more abundant, particularly at Site 3, the wide confidence intervals and non-significant *p*-values indicated insufficient statistical support (Appendix A).

In sediment samples, seasonal patterns were also evident. The *intl-1* gene showed higher levels during the 2021 wet season (BF = 69.07), particularly at Site 3, but lower concentrations during the 2022 wet season (*p* = 0.025). The *tetQ* gene showed elevated levels in both campaigns (BF = 37.56), especially in the 2022 wet season across all sites. In contrast, *sul-1* was markedly lower during the 2022 wet season (BF = 1090), particularly at Sites 2 and 3, while *tetA* was higher (BF = 474.57), mainly at Site 2. The *sul-2* gene showed reduced levels (BF = 78.36), and *qnrS* was relatively elevated (BF = 3.14). Site-specific differences are illustrated in Figure 1.

No seasonal differences were supported for *ermB* (BF = 1.15) and *dfrA12* (BF = 0.71), as their low BF values and confidence intervals included zero (Appendix A). Nonetheless, Figure 1 shows that *ermB* maintained relatively high abundance across all sediment samples, particularly at Sites 1 and 3.

### 2.3. Bacteria Diversity Analysis

A total of 11,477 taxa, representing 1966 bacterial genera, were identified in the chrimosomal dataset. Of these, 251 genera (4044 taxa; 35%) were classified as potential human pathogens. In contrast, the plasmidic dataset (the plasmidome) comprised 472 taxa from 233 genera, 70 of which (207 taxa; 44%) were associated with potential pathogenicity. This indicates a higher relative proportion of pathogenic taxa within the plasmidome.

Figure 2 shows the taxonomic composition of bacterial genera identified in the chromosomal and plasmid fractions of sediment metagenomes. Chromosomal profiles were dominated by environmentally associated genera, such as *Acidovorax*, *Acinetobacter*, and *Pseudomonas*, reflecting the native microbial community’s structure. In contrast, the plasmid-associated fraction exhibited a distinct taxonomic composition, with a higher relative abundance of clinically relevant genera, including *Enterobacter*, *Klebsiella*, *Providencia*, and *Leclercia*.

As shown in Figure 3, alpha diversity analyses of potential pathogenic genera revealed that species richness tended to increase with contamination level, especially in the plasmid-associated dataset. Additionally, the Shannon diversity index revealed that plasmid-derived profiles from Site 3 consistently exhibited the highest diversity across all sampling campaigns. These results suggest that both richness and evenness of plasmid-borne pathogenic genera increase under more polluted conditions, whereas chromosomal diversity remained comparatively stable.

To assess differences in community composition, beta diversity was evaluated using principal coordinates analysis (PCoA) based on Bray–Curtis dissimilarities from CLR-transformed taxonomic profiles (Figure 4). In the chromosomal dataset (Panel A), samples clustered by both site and sampling campaign, reflecting strong spatiotemporal structuring. PERMANOVA confirmed significant variation across sites (R^2^ = 0.30, F = 5.10, *p* = 0.001) and campaigns (R^2^ = 0.17, F = 2.42, *p* = 0.001).

The plasmid dataset (Panel B) exhibited lower overall variability; however, Site 1 (the least contaminated) formed a more distinct cluster compared to Sites 2 and 3. PERMANOVA also indicated significant differences by site (R^2^ = 0.33, F = 6.04, *p* = 0.001) and campaign (R^2^ = 0.14, F = 1.93, *p* = 0.02). Samples from the two wet-season campaigns (Sampling 1 and 3) showed more similarity in the chromosomal dataset, suggesting seasonal influence.

### 2.4. Abundance of Potential Pathogens

Spatial variation had a greater influence than seasonal changes on the distribution of potentially pathogenic bacterial species in both chromosomal and plasmid fractions. In the chromosomal dataset, Site 3 (the most contaminated) consistently showed higher abundances of several pathogenic species. For instance, during the 2022 dry season, *Acinetobacter johnsonii* (H = 7.73, *p* = 0.021) and *A. parvus* (H = 17.41, *p* < 0.001) were more abundant. In the 2022 wet season (Sampling 3), elevated levels were observed for *Aeromonas caviae* (H = 19.20, *p* < 0.001), *Moraxella osloensis* (H = 16.71, *p* = 0.0002), and *Pseudomonas alcaligenes* (H = 18.11, *p* < 0.001). *Cutibacterium acnes* (H = 13.78, *p* = 0.0010) and *Enterobacter cloacae* (H = 23.29, *p* < 0.001) were consistently enriched at Site 3 across all campaigns.

In the plasmid-associated fraction, several species also displayed increased abundance at Site 3. Notably, *Leclercia adecarboxylata* (H = 16.06, *p* < 0.001), *Pseudomonas putida* (H = 15.76, *p* = 0.0004), and *Moraxella osloensis* (H = 17.02, *p* = 0.0002) were more abundant during the 2022 wet season. Additional enriched species included *Escherichia fergusonii* (H = 15.75, *p* = 0.0004), *Microbacterium paraoxydans* (H = 13.20, *p* = 0.0014), *Proteus vulgaris* (H = 11.66, *p* = 0.0029), and *Providencia rettgeri* (H = 17.69, *p* < 0.001). However, post hoc tests did not consistently confirm these trends.

### 2.5. Abundance of Distinctive ARGs in Metagenomic Data

Sixty-five ARGs were identified across the contigs in all samples, and were grouped into 16 resistance categories based on their corresponding drug classes as defined by the CARD database. Figure 5 illustrates the normalized abundance (RPKM) of each ARG category, organized by sampling site and campaign, providing insight into distribution patterns.

Aminoglycoside resistance showed significant spatial variation across sites (H = 18.5, *p* = 9.60 × 10^−5^). Post hoc analyses revealed significantly lower resistance levels at Site 1 compared to Site 3 (Z = −4.02, *p.adj* = 0.00017) and Site 2 compared to Site 3 (Z = −3.33, *p.adj* = 0.0026).

Similarly, beta-lactam resistance also showed significant site-level variation (H = 13.1, *p* = 4.20 × 10^−4^), with significantly lower levels observed at Site 1 versus Site 3 (Z = −3.38, *p.adj* = 0.0021) and Site 2 versus Site 3 (Z = −3.44, *p.adj* = 0.0017). Macrolide resistance demonstrated clear spatial differences (H = 15.8, *p* = 7.25 × 10^−5^), with lower levels in Site 1 (Z = −3.67, *p.adj* = 0.00073) and Site 2 (Z = −3.89, *p.adj* = 0.00031) compared to Site 3. Plasmid-associated ARGs in these categories showed no significant spatial differences.

Kruskal–Wallis tests revealed significant differences in chromosomal ARGs to colistin, diaminopyrimidine, fluoroquinolone, rifamycin, lincosamide, and phenicol (H = 6.51–6.52, *p* = 0.039 for each). Although none of the pairwise contrasts remained significant after Bonferroni correction (all *p.adj* > 0.05), unadjusted *p*-values indicated potential seasonal trends in chromosomal-ARG abundance. For example, higher levels were observed in the 2021 wet season compared to the 2022 dry and wet seasons for colistin, fluoroquinolone, and phenicol; whereas diaminopyrimidine and rifamycin levels were lower in the 2022 dry season.

In contrast, plasmid-associated beta-lactam (H = 10.23, *p* = 0.0159), macrolide (H = 10.09, *p* = 0.0159), macrolide-streptogramin (H = 9.74, *p* = 0.0159), and MLS (H = 9.88, *p* = 0.0159) resistance were exclusively detected at Site 3 across all sampling campaigns and showed spatial variation. A significantly higher abundance of ARGs in plasmidic contigs was observed in the 2022 wet season compared to the 2021 wet season (H = 8.79, *p.adj* = 0.0251) and in the 2022 dry season compared to the 2021 wet season (H = 7.62, *p.adj* = 0.0483), with no significant differences between the 2022 dry and wet seasons.

Appendix A provides a comprehensive summary of ARGs, including mapped reads per sample, reference accession numbers, associated resistance phenotypes, coverage identity, and contig taxonomy.

## 3. Discussion

This study provides evidence that, although antibiotic resistance and potential pathogenic bacteria vary significantly across a pollution gradient in a tropical urban river, distinct patterns are observed depending on whether the sequences are chromosomal or plasmid-associated. Chromosomal ARGs exhibited marked spatial differences, with significantly higher abundance at the most contaminated site (Site 3), particularly for ARGs against aminoglycosides, beta-lactams, and macrolide antibiotics. In contrast, plasmid-associated ARGs exhibited significant temporal variation, with higher abundance detected during the middle of the 2022 wet season compared to the end of the 2021 wet season. These findings underscore the importance of distinguishing between chromosomal and plasmid-derived sequences when evaluating environmental resistomes. Recent work [7] has emphasized this distinction by demonstrating contrasting diversity patterns between chromosomal and plasmid profiles in this polluted tropical river. Moreover, the present results highlight how methodological choices, such as the type of matrix analyzed (water, sediment), sequencing strategy (qPCR or metagenomics), genomic compartment (chromosome or mobile genetic elements), and sampling timing, can influence resistance surveillance outcomes and interpretations.

### 3.1. Impacts of Pollution on the Surface Water of the Virilla River Watershed

Across the three sampling campaigns, a clear pollution gradient was observed in the surface waters of the Virilla River. Fecal indicator bacteria, including fecal coliforms and *E. coli*, reached the highest concentrations at sites heavily impacted by anthropogenic activities and wastewater discharges, particularly at Site 3. This pattern mirrors trends observed in other urban rivers worldwide, where untreated or insufficiently treated wastewater is commonly discharged due to aging infrastructure or rapid urban expansion [17]. In Latin America, for example, less than 60% of wastewater is treated before release into natural water bodies, exacerbating contamination in densely populated watersheds [17]

In Costa Rica, urbanization has markedly impacted water quality in the Virilla River. Wastewater discharges from urban areas are associated with elevated concentrations of pollutants, including biochemical oxygen demand (BOD), chemical oxygen demand (COD), ammonium, and nitrites. These pollutants tend to accumulate as urbanization intensifies, in contrast to upstream regions dominated by agricultural or forested land use [18]. Notably, Site 3 is located downstream of the San José Wastewater Treatment Plant (SJ-WWTP) and receives inputs from densely populated residential and industrial zones, including multiple public and private healthcare facilities. These point sources likely explain the elevated levels of multidrug-resistant bacteria and plasmid-associated consistently reported at this location [2,7]. Moreover, microbial contamination poses a serious health risk, as high levels of fecal indicator bacteria and *E. coli* virulence and resistance genes have also been documented, particularly at sites influenced by human activity [1].

Surface water quality in the Virilla River showed clear spatial and seasonal variation, with elevated contamination levels detected during both the dry and wet seasons of 2022, particularly at downstream sites influenced by urban pollution. Reduced rainfall and limited runoff during the dry season likely favored pollutant accumulation, while intense precipitation during the wet season may have mobilized contaminants from informal settlements, landfills, and impervious urban surfaces, further impairing water quality. Previous research in this watershed [1] as well as in other tropical river systems [11] has documented comparable seasonal dynamics, including significantly higher *E. coli* concentrations during wet periods. This pattern likely reflects increased microbial transport and pollutant loading during high-flow conditions. In the context of climate change, these processes may be intensified, as projected increases in rainfall variability could enhance contaminant mobilization during extreme precipitation events or promote pollutant concentration during periods of drought [12].

### 3.2. Seasonal Dynamics of ARGs Quantification

The distribution of ARGs in aquatic ecosystems is increasingly recognized as a dynamic process shaped by seasonal hydrological variability, anthropogenic pollution, and microbial community structure [8,11,19,20]. In the Virilla River, seasonal shifts in both waterborne and sediment-associated ARGs reflect hydrological processes such as dilution, accumulation in sediments, and mobilization driven by runoff [7,8,21]. Physicochemical conditions and land-use patterns appear to play a more decisive role than individual gene dynamics in determining the spatiotemporal distribution of ARGs. These observations are consistent with findings from tropical rivers, where seasonal runoff from urban and agricultural sources strongly influences ARG transport and persistence [8,21,22], and sediments serve as key reservoirs during high-flow events [22].

Remarkably, all sampling campaigns were conducted during La Niña conditions, which are typically associated with increased rainfall in Central America. ERA5-Land data (Appendix A) confirmed that precipitation levels during the 2022 wet season (Sampling 3) were higher than in 2021. These conditions likely enhanced runoff and mobilized ARGs, pathogenic bacteria, and antibiotic residues from diffuse pollution sources, including informal settlements, agricultural areas, and poorly managed landfills. Although modeling these climatic anomalies was beyond the scope of this study, their hydrological impacts, particularly increased dilution, sediment resuspension, and pollutant input, may have contributed to the temporal variability observed in ARG profiles [12]. Future research should consider the role of interannual climate variability and ENSO phase transitions (e.g., La Niña to El Niño) to better distinguish seasonal from year-specific influences.

To provide a reference for interpreting ARG patterns, Site 1, characterized by lower urban pressure and predominantly forested and agricultural land use, consistently exhibited lower ARG abundance and diversity, consistent with values in minimally impacted freshwater systems [11,23]. In contrast, Sites 2 and 3, which are more exposed to urban, industrial, and healthcare-related discharges, showed a marked enrichment of genes such as *intI-1*, *bla_TEM_*, and *qnrS*, particularly in the sediment samples. These trends reflect a significant anthropogenic influence throughout the gradient.

Agricultural activity in upstream areas may further contribute to the dissemination of ARGs, particularly those encoding resistance to sulfonamides and tetracyclines. In Costa Rica, tetracyclines (e.g., oxytetracycline, chlortetracycline) are widely used in tomato farming and livestock production, with residues detected in soils and nearby rivers [24,25]. Seasonal manure application during the rainy season may promote the mobilization of *tet(Q)* and *sul* genes into sediments [10,24]. In parallel, treated hospital and domestic wastewater may serve as point sources for sulfonamide resistance genes (*sul1*, *sul2*) transported via resistant bacteria [26,27,28]. The higher abundance of ARGs at contaminated sites during the dry season likely reflects pollutant concentration under low-flow conditions, particularly near wastewater treatment plants (WWTPs).

National statistics confirm that antimicrobial use remains high across sectors: in 2019, Costa Rica reported 14.32 DDD/1000 inhabitants/day in human healthcare, with higher consumption in the public sector [29], while 71.8% of tomato producers reported the use of bactericides or antibiotics [25]. The widespread use of tetracyclines in agriculture, aquaculture, and livestock, often lacking proper regulation, may exacerbate environmental contamination. Although the National Action Plan Against AMR (2018–2025) has improved surveillance in the public health sector, additional regulatory measures are necessary, especially in the private healthcare and agricultural sectors [30].

Water and sediment host distinct microbial communities, influencing ARGs’ behavior. During rainy periods, sediment conditions may promote the persistence of tetracycline resistance genes (*tetM*, *tetQ*, *tetW*) in biofilms and particle-bound states [31]. Warmer temperatures during the dry season may, in turn, enhance horizontal gene transfer in the water column due to increased microbial activity [21]. The contrasting behavior of *intI-1*, declining in water but increasing in sediments, highlights the role of sediment biofilms in stabilizing mobile genetic elements during hydrological disturbance. Sediments at Site 3, which consistently showed elevated *intI-1*, may represent a hotspot for gene exchange due to organic matter accumulation.

Beyond acting as passive recipients, sediments function as long-term reservoirs of ARGs, with the potential to release contaminants (and microbes) during resuspension events. Flooding associated with climate change may exacerbate this risk by remobilizing previously trapped ARGs into the water column [32]. Experimental studies have confirmed that genes such as *bla_TEM_* and *sul1* can persist in riverbed sediments for extended periods, even under dry conditions [33]. Metagenomic surveys of peri-urban rivers further support this role, demonstrating accumulation of ARGs along anthropogenic gradients, often co-localized with class 1 integrons [34].

These findings emphasize the importance of including sediments in AMR surveillance efforts and of monitoring chromosomal as well as plasmid-associated ARGs. The consistent enrichment of resistance genes under contrasting seasonal conditions highlights the complex interactions between land use, climate, and microbial dynamics in shaping environmental resistomes.

### 3.3. Spatial Patterns Drive Microbial Community Composition and ARG Distribution More than Temporal Variability

This study shows contrasting responses between chromosomal and plasmidome fractions to environmental contamination, with clear spatial patterns that exert a stronger influence than seasonal variability, particularly at the level of plasmidic contigs. To avoid confusion, we distinguish between the plasmidome, defined as the entire set of plasmid-derived sequences, and plasmid-associated ARGs, which refer only to the ARGs located on those sequences.

At the most contaminated site (Site 3), plasmidome richness increased substantially. Specifically, while plasmid-associated diversity rose with contamination, chromosomal diversity remained relatively stable. This stability, even under highly polluted conditions, may reflect the inherent resilience of core microbial communities, a phenomenon previously reported in heavily impacted coastal environments [35]. The persistent enrichment of plasmid-associated ARGs and diversity at the most polluted sites, regardless of season, suggests that plasmids function as stable genetic reservoirs for these ARGs, shaped primarily by anthropogenic pressures rather than short-term environmental fluctuations. This contrasts with the more variable patterns observed in chromosomal ARGs, underscoring the potential of plasmids to reflect long-term selective pressures in disturbed environments. The plasmidome appears to support microbial adaptation by sustaining the carriage and dissemination of resistance traits [3,36]. This ecological role aligns with emerging models that consider plasmids not only as vehicles for HGT, but also as key contributors to microbial community structure and functional potential under persistent pollution [15,36].

Taxonomic profiling of sediment metagenomes revealed a marked separation between chromosomal and plasmid-associated bacterial communities. Chromosomal fractions were dominated by environmentally adapted genera such as *Acidovorax*, *Acinetobacter*, and *Pseudomonas*, while the plasmidome showed higher representation of clinically relevant genera, including *Enterobacter*, *Klebsiella*, *Providencia*, and *Leclercia.* This contrast underscores the plasmidome’s ecological specialization in harboring mobile elements associated with resistance and virulence.

Strong spatial patterns in microbial composition were also detected, with chromosomal communities showing distinct spatial clustering. This structure is consistent with previous findings from urban rivers, where point-source pollution from industrial and agricultural sources shapes localized microbial assemblages [11,37]. In contrast, plasmid-associated communities displayed lower spatial variability, except at Site 1 (the least contaminated), which differed markedly from Sites 2 and 3. A similar pattern has been reported for plasmid-associated ARGs and MGEs in polluted European rivers, such as the Ter River in Catalonia, where downstream sites receiving wastewater show increased enrichment and homogenization of these elements, likely due to horizontal gene transfer [38]. This suggests that contamination may exert a homogenizing effect on the plasmidome, reducing spatial differentiation.

Several opportunistic pathogens were consistently enriched at the most contaminated site, highlighting the potential health risks associated with polluted urban rivers. *Acinetobacter johnsonii* and *A. parvus* are known for their environmental persistence and their role in healthcare-associated infections, especially in patients with indwelling devices [39,40]. *Aeromonas caviae*, a common aquatic bacterium, has been linked to gastrointestinal and bloodstream infections in immunocompromised individuals [41]. *Cutibacterium acnes*, although typically a skin commensal, is increasingly associated with biofilm-related infections on medical implants [42].

Other enriched pathogens included *Enterobacter cloacae*, *Escherichia fergusonii*, and *Proteus vulgaris*, all members of the Enterobacteriaceae family, which are frequently implicated in urinary and bloodstream infections. These species often harbor multidrug resistance mechanisms and have been isolated from environmental sources, including water [43]. *Leclercia adecarboxylata*, though relatively rare, has emerged as a cause of polymicrobial infections, including urinary and respiratory tract infections, even in immunocompetent individuals [44]. *Microbacterium paraoxzayetydans* has been reported in catheter-related bloodstream infections, particularly among chronically ill patients [45].

Additional enriched taxa included *Moraxella osloensis*, associated with respiratory infections in individuals with underlying conditions [46], and *Pseudomonas putida*. This widespread environmental bacterium can act as a pathogen in aquaculture and clinical settings. It also carries virulence and antibiotic resistance genes [47]. These findings underscore the role of urban rivers as reservoirs for emerging and opportunistic pathogens, highlighting the importance of environmental monitoring in public health strategies.

Contig-based analysis revealed distinct spatial trends in chromosomal ARGs, with significantly elevated resistance levels at Site 3, particularly for aminoglycosides, beta-lactams, and macrolides. This indicates that chronic pollution and environmental selection contribute to the persistence of chromosomal resistance. In contrast, plasmid-associated ARGs were more abundant during the rainy season (wet seasons 2021 and 2022). Resistance categories, such as beta-lactam, macrolide, and MLS, were especially enriched during these periods, suggesting increased inputs of antibiotics and other selective agents through runoff and wastewater discharge.

Beyond spatial and seasonal variations, the distribution of ARGs can also be influenced by physico-chemical parameters, including temperature, pH, and dissolved oxygen (DO). These variables can modulate microbial composition and the abundance of MGEs, thereby indirectly shaping resistance dynamics [21,22,48]. Nonetheless, as highlighted in our previous work, anthropogenic contamination is the main driver of ARG dynamics in the Virilla River, acting as a strong selective force that promotes the evolution and dissemination of antibiotic resistance [7].

Pollution’s influence becomes particularly evident when comparing chromosomal and plasmid-associated ARGs. Plasmid-associated ARGs exhibited greater fluctuations, likely due to the dynamic nature of plasmids, which can be rapidly acquired or lost in response to environmental stressors. Episodic selection events, such as rainfall-driven contaminant pulses, may trigger sharp increases in plasmid-mediated resistance. In contrast, chromosomal ARGs tend to persist more consistently, even in the absence of immediate selective pressures.

These findings emphasize the need for site-specific interventions, particularly at urban discharge points (Site 3) and agricultural runoff zones (Site 2), to reduce ARG inputs and control their spread. Monitoring programs embedded within a One Health framework would facilitate early detection and improved management of antimicrobial resistance in urban aquatic systems.

Finally, the consistent enrichment of plasmid-associated ARGs and potentially pathogenic taxa at Site 3 underlines the utility of plasmids as sensitive biomarkers of urban contamination. Their capacity to respond rapidly to environmental pressures and to carry multiple resistance genes makes them valuable indicators of pollution-driven microbial shifts. Plasmid profiling in metagenomic studies thus offers a powerful early-warning tool for detecting anthropogenic impacts in freshwater ecosystems affected by wastewater discharge and urban runoff.

#### Limitations of the Study

A key limitation is the uneven seasonal sampling, comprising two wet season campaigns (2021 and 2022) and only one during the dry season (2022). This imbalance resulted from logistical constraints and the inclusion of a previously collected dataset from late 2021, which coincided with the wet-to-dry seasonal transition. Although this design allowed the identification of seasonal and intra-seasonal trends within a wet-to-wet cycle, future studies should implement more balanced sampling across seasons to enhance temporal resolution and comparability.

## 4. Materials and Methods

### 4.1. Study Area and Sampling Design

This study was conducted along the main course of the Virilla River watershed, located in the western Central Valley of Costa Rica (Figure 6). This watershed, part of the larger Grande de Tárcoles River Basin, has been heavily impacted by urbanization, with untreated sewage and industrial discharges contributing to high pollution levels [17,18].

Three sampling sites were selected along an altitudinal and pollution gradient. At each site, water and sediment samples were collected during three sampling campaigns: November 2021 (wet season 2021), April 2022 (dry season 2022), and August 2022 (wet season 2022). Triplicate samples of both matrices were obtained per site and campaign, totalling 18 samples per campaign and 54 samples overall (27 water samples and 27 sediment samples).

The main characteristics of each site -including coordinates, elevation, dominant land use, and analysis performed- are detailed in Appendix A. During the study period, Costa Rica was under “La Niña” conditions (cold ENSO phase). The 2022 wet season received slightly more rainfall than 2021, but both were markedly wetter than the 2022 dry season. Daily precipitation data are provided in Appendix A.

Although the study aimed to assess seasonal variation, the sampling design included three campaigns to capture a broader range of hydrological conditions. The wet season 2021 campaign, although conducted before the others, took place during the late rainy season, near the transition to the dry period. Its inclusion allowed the incorporation of high-quality data collected and extended the temporal scope of the study. Moreover, climatological records (see Appendix A) indicate that rainfall patterns during this period differed from those observed in the 2022 wet season, thus representing distinct hydrological contexts relevant for the interpretation of ARG dynamics.

### 4.2. Microbiological and Physical-Chemical Analysis

Water samples were collected in sterile 1-liter containers, and sediment samples were gathered from the river shore at a depth of ~30 cm using a sterile spoon and transferred to sterile plastic bags. All samples were double-bagged and transported in insulated coolers with ice packs to preserve cold chain conditions. Upon arrival at the INISA-UCR laboratory, the water sample temperature was confirmed to be within the range of 4 °C to 8 °C. Samples were processed within 20 h of collection.

Microbiological and physico-chemical parameters were measured only in water samples. Fecal coliforms, *Escherichia coli* (*E. coli*), and *Enterococcus faecalis* (*E. faecalis*) were quantified using the most probable number technique (Standard Methods for the Examination of Water and Wastewater, 9221E, 9221F, and 9230B; Part 9000. Microbiological Examination. 23rd ed.; American Public Health Association: Washington, DC, USA, 2017) [49]. Oxygen saturation, conductivity, pH, and temperature were analyzed using a multiparameter probe (YSI model 85A, Yellow Springs, Greene, OH, USA). Turbidity was examined at the Health Research Institute (INISA) laboratory using the Standard Method 2510B [49].

The National Sanitation Foundation Water Quality Index (NSF-WQI) was calculated to assess overall water quality at each sampling site. In this study, the index was based on five key parameters: dissolved oxygen (DO), fecal coliforms, pH, temperature, and turbidity. Each parameter was converted into a sub-index using NSF rating curves, and a weighted average was calculated. The following weights were applied to reflect their relative importance: DO (0.31), fecal coliforms (0.30), pH (0.14), temperature (0.13), and turbidity (0.12). Although the original NSF-WQI includes nine parameters, previous studies have validated reduced versions under data constraints, provided the weights are proportionally adjusted and reported. This adapted approach has been applied in various studies under similar constraints [50,51,52]. All calculations were performed using standard Excel-based methods and verified manually. By integrating these variables, the NSFQI provides a comprehensive assessment of water quality, enabling the detection of potential anthropogenic contamination. It further classifies surface waters into five distinct quality categories: very good (90.1–100), good (70.1–90), fair (50.1–70), poor (25.1–50), and very poor (0–25) [50,53,54].

### 4.3. DNA Extraction and Quantification

Water samples were prefiltered using a filtration device (Sartorius^®^, Göttingen, Germany) equipped with an 80-micrometer glass fiber prefilter (13,400–47-Q, Sartorius^®^, Germany) to remove larger particles. Subsequently, prefilter water was filtered through a 0.22-μm cellulose nitrate filter membrane (47-millimeter diameter; 11,327–47–N, Sartorius^®^, Germany).

According to the manufacturer’s protocol, DNA was extracted from both water filters and sediment samples using DNeasy PowerSoil Pro (Qiagen, Venlo, The Netherlands). For each sample, two filters were processed (250 mL per filter, totaling 500 mL) to ensure sufficient DNA yield. Similarly, two separate extractions were performed per sediment sample using 250 mg of sediment each, and the resulting DNA was pooled for downstream analyses. DNA concentration and purity were assessed using a NanoDrop^TM^ 2000 spectrophotometer (ThermoFisher Scientific, Waltham, MA, USA). Water samples yielded between 7.5 and 50 ng/µL (A260/280 values: 1.6–2.1), while sediment extracts ranged from 7.0 to 59 ng/µL (A260/280 values: 1.6–1.9). Extracts were aliquoted and stored at −80 °C until further processing, including ARGs quantification and sequencing. Although only 250 mL was filtered per membrane, this volume consistently provided DNA of adequate yield and quality for qPCR quantification and Illumina NovaSeq metagenomic sequencing. This approach has been validated in previous studies within the same river system [7].

### 4.4. Quantification of Antibiotic Resistance Genes (ARGs)

Quantitative PCR (qPCR) was used to quantify selected ARGs in water and sediment samples. The targeted genes conferred resistance to sulfonamides (*sul-1* and *sul-2*), tetracyclines (*tet(A)* and *tet(Q)*), beta-lactams (*bla_CTX-M_*, *bla_OXA_*, and *bla_TEM_*), macrolides (*ermB*), quinolones (*qnrS*), and trimethoprim (*dfrA12*). Detailed qPCR conditions, including primer sequences, amplicon sizes (85 to 190 bp), primer concentrations (200–600 nM), and annealing temperatures (79.8–90.02 °C), are provided in Appendix A, following Barrantes-Jiménez et al. [7]. ARG abundances were normalized to *16S rRNA* gene copies to account for differences in bacterial biomass. The April 2022 data, previously published [7], are integrated here with two additional sampling campaigns to provide a broader temporal and spatial perspective on ARG dynamics.

### 4.5. Shotgun Metagenomic Sequencing and Quality Control

DNA sample extracts were frozen and shipped to Novogene Inc. (Sacramento, CA, USA) for metagenomic library preparation and 150 bp paired-end sequencing on an Illumina NovaSeq 6000 platform, yielding approximately 2 × 2 GB of data per sample. Libraries were prepared using the ABclonal Rapid Plus DNA Library Kit (ABClonal, Woburn, MA, USA). Quality control included concentration assessment with a Qubit 2.0 fluorometer (Invitrogen, Carlsbad, CA, USA) and insert size verification with an Agilent 2100 Bioanalyzer (Agilent, Santa Clara, CA, USA).

Adapter removal, quality trimming, and filtering of raw reads were performed using fastp v0.20.1 [55], with default parameters except for -e value set to 25. Reads mapping to the human genome (hg19) were identified and removed using Bowtie2 v2.5.0 [56] with the very-sensitive-local alignment mode.

### 4.6. Taxonomic Assignment and Diversity Analysis

Taxonomic classifications of filtered reads were performed at the species level using Kraken2 v 2.1.2 [57] with two previously indexed custom databases and a classification threshold value of 0.5. One database included chromosomal sequences from GTDB r207 v2 [58], while the other contained only plasmid sequences from the same source. This approach enabled the generation of two distinct datasets: chromosomal reads and plasmidic reads for downstream analyses [7].

Taxonomic reports generated by Kraken2 were processed using Bracken v2.8 [59] to calculate read counts for each taxon. The resulting matrix was imported into R as a phyloseq object using the bracken to phyloseq (b2p) in-house package [60]. Full taxonomy paths (from phylum to species) were assigned based on the NCBI taxonomy identifiers provided by Kraken2, utilizing the taxonomizr package in R. Detailed instructions for this step can be found in the repository: https://github.com/braddmg/b2p, accessed on 6 February 2025.

For microbial diversity analyses, chromosomal and plasmidic read datasets were processed separately. Samples were rarefied to the minimum read count observed across all samples within each dataset. Bar plots showing the most abundant genera were generated using the phyloseq (Bioconductor 3.20): 1.50.0 and microViz 0.12.6 R packages (https://joss.theoj.org/papers/10.21105/joss.03201.pdf, accessed on 6 February 2025).

To evaluate the presence of potentially pathogenic bacteria, we used the curated list of human bacterial pathogens compiled by Bartlett et al. [61], which includes over 4000 clinically relevant taxa. Due to the limited resolution of species-level taxonomic classification using short-read metagenomic data (150 bp), we adopted a conservative strategy by screening at the genus level. All taxa assigned to genera that include at least one human pathogenic species in the Bartlett list were retained, resulting in 4044 taxa across 251 genera. These were used to assess the alpha and beta diversity analysis in chromosomal and plasmid fractions. Among the 20 most abundant species identified in sediment samples, 13 were classified as potential human pathogens. Kruskal–Wallis tests followed by Dunn’s post hoc tests were performed exclusively on these 13 pathogenic species to evaluate differences in abundance across sampling sites and campaigns. The complete list of pathogenic genera is provided in Appendix A.

Observed richness and the Shannon diversity index were calculated to assess alpha diversity in each sample. For beta diversity, taxonomic data were transformed using centered log-ratio (CLR) normalization and visualized through principal coordinates analysis (PCoA), grouping samples by sample type, site, and date. Permutational analysis of variance (PERMANOVA) was applied to evaluate differences in the composition of potential pathogenic taxa across sites.

### 4.7. ARGs Detection from Sediment Samples

Filtered reads from sediment samples collected at the same site and sampling event were co-assembled using MEGAHIT [62] with k-mer sizes of 33, 55, 77, 99, and 127. Resulting contigs were mapped to their corresponding FASTQ files using Bowtie2 v2.5.0 [55]. Contigs shorter than 500 bp were removed, and assembly quality was assessed using MetaQUAST v2.2 [63].

Assembled contigs from each co-assembly were analyzed using the PlasX pipeline [15]. Contigs were first imported into Anvio v8.0 [64], where coding sequences (CDSs) were identified with Prodigal [65]. Annotations were performed using the COG v14 and Pfam v33.1 databases [66,67] to generate de novo protein families. The annotations were then compared against a pre-trained plasmid database, and PlasX assigned a score to each contig. Contigs with scores > 0.5 were classified as plasmidic; those with lower scores were classified as chromosomic.

ARGs were annotated in the contigs using Abricate v1.0.1 (https://github.com/tseemann/abricate, accessed on 6 February 2025) with the CARD database [68], applying an identity threshold of 80% and a minimum coverage of 80%. Coordinates of each annotated ARG were mapped to the corresponding FASTQ files using pysam and Samtools [69]. The number of reads aligning to each ARG region was counted, and read counts were normalized by calculating RPKM (reads per kilobase per million mapped reads) values, dividing the raw counts by the region length (in kilobases) and the total number of mapped reads. These normalized values were used in downstream statistical analyses.

Contigs containing ARGs were taxonomically classified using Kaiju with the Swiss-Prot protein database [70]. This analysis provided insight into the taxonomic origins of ARGs within the samples. All scripts are available at https://gitlab.com/legema.inisa/Metagenomics (accessed on 6 February 2025).

### 4.8. Precipitation Data

Precipitation data for the three sampling sites across all campaigns were obtained from the ERA5-Land reanalysis dataset, developed by the European Centre for Medium-Range Weather Forecasts (ECMWF) as part of the Copernicus Climate Change Service (C3S) (see Appendix A). ERA5-Land provides high-resolution (11.132 km per pixel) terrestrial climate variables derived from observational data and model assimilation. Data distribution was assessed using the Shapiro–Wilk test, which indicated non-normality. Consequently, a non-parametric Kruskal–Wallis test was applied to evaluate significant differences in precipitation between seasons and sites (*p* < 0.001), followed by Dunn’s post hoc comparisons with Holm correction. The analyses showed significantly lower precipitation levels during the dry season compared to both wet seasons.

### 4.9. Data Analysis and Visualization

Data normality was first evaluated using the Shapiro–Wilk test for each variable, site, and sampling campaign, with physicochemical and microbiological parameters, ARG values, and bacterial abundance. Due to inconsistent normality across results, the non-parametric Kruskal–Wallis test was used to assess significant differences. When significant effects were detected (*p* < 0.05), Dunn’s post hoc test with Bonferroni correction was applied to identify specific site-level differences.

Heatmaps were generated to visualize the geometric mean of ARG relative abundances detected by qPCR across sites and sampling periods in both water and sediment matrices. ARGs’ relative abundances were log-transformed to address non-normal distributions, and the geometric mean was used to minimize the influence of extreme values. A logarithmic color scale from 0.00001 to 10 copies/L was applied for visualization.

Tobit regression models were used to evaluate the effect of sampling on ARGs values detected by qPCR in each matrix (water and sediment). These models were selected for their capacity to handle censored data in the response variable [71], allowing robust inference when values fall below detection limits. To control potential confounders, both unadjusted and site-adjusted models were tested.

Additionally, Bayes factors were calculated for each model to strengthen the evaluation of evidence supporting the null and alternative hypotheses. A Bayes factor greater than 1 supports the alternative hypothesis, while values below 1 support the null hypothesis [72]. For all models, β coefficients, 95% confidence intervals (CIs), t-values, *p*-values, and Bayes factors were reported to assess the magnitude, precision, and significance of observed relationships.

This comprehensive dual-model approach allowed the evaluation of site, sampling campaigns, and matrix effects on the relative abundance of ARGs and diversity indices in chromosomal and plasmid-associated potential pathogens.

All statistical analyses were conducted in R (version 4.3.0,). Kruskal–Wallis and Dunn’s post hoc test with Bonferroni adjustment for multiple comparisons was performed using FSA, dplyr, and rstatix packages. Tobit regression models were fitted using the VGAM package, linear models were implemented with the lm() function, and Bayesian inference was carried out with the bayestestR package. Additional libraries, including tidyverse, parameters, MASS, and scales, were used for data processing, diagnostics, and visualization.

## 5. Conclusions

Seasonality and pollution gradients had a significant influence on the abundance and distribution of antibiotic resistance genes (ARGs), bacterial pathogens, and plasmid composition in this tropical urban river. Quantitative PCR analysis revealed a consistent enrichment of ARGs in sediments, particularly during the wet seasons, likely due to the resuspension of particle-bound bacteria and enhanced gene mobilization, patterns that align with previous studies in tropical freshwater systems of other (sub)tropical regions [73].

The plasmidome composition differed markedly between sites with distinct pollution levels, with a clearer separation between the least and most contaminated locations. This contrasts with chromosomal profiles, which showed more variability across seasons. These findings support previous work [7], suggesting that plasmids are sensitive indicators of anthropogenic influence and may play a central role in the dissemination of resistance under environmental stress.

Sediments emerged as important reservoirs and potential sources of ARGs and pathogens, reinforcing the need to include this compartment in AMR surveillance. Moreover, the enrichment of specific ARGs in plasmids and their variation across sites suggest that mobile genetic elements should be explicitly monitored in environmental AMR studies.

Combining qPCR and metagenomics provides a powerful approach to capture the ecological complexity of AMR dynamics in tropical aquatic ecosystems. These insights underscore the importance of incorporating sediment analyses, plasmid-level resolution, and seasonal sampling strategies into environmental AMR monitoring frameworks, particularly in regions with rapid urban expansion and limited wastewater treatment coverage.

Incorporating plasmid-associated ARGs into AMR surveillance frameworks and identifying site-specific and seasonal resistance hotspots can enhance early detection of environmental contamination, guide land-use and wastewater management strategies, and strengthen public health preparedness.

## Figures and Tables

**Figure 1 antibiotics-14-00798-f001:**
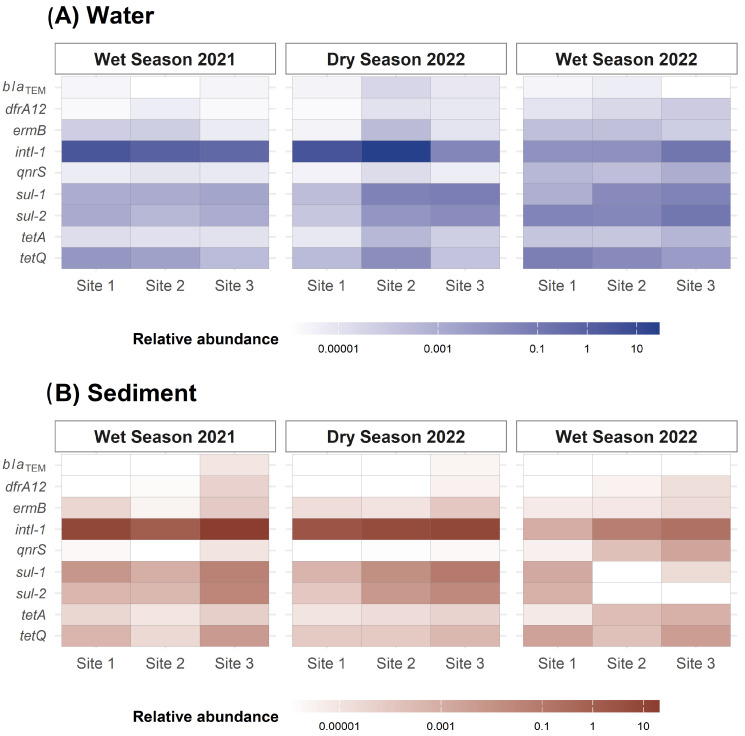
Relative abundance of ARGs in (**A**) water and (**B**) sediment samples from the Virilla River by qPCR during the three sampling campaigns. Values represent ARG copies normalized to *16S rRNA* gene copies (i.e., copies per *16S rRNA* gene). The color scale is logarithmic and ranges from 0.00001 to 10.

**Figure 2 antibiotics-14-00798-f002:**
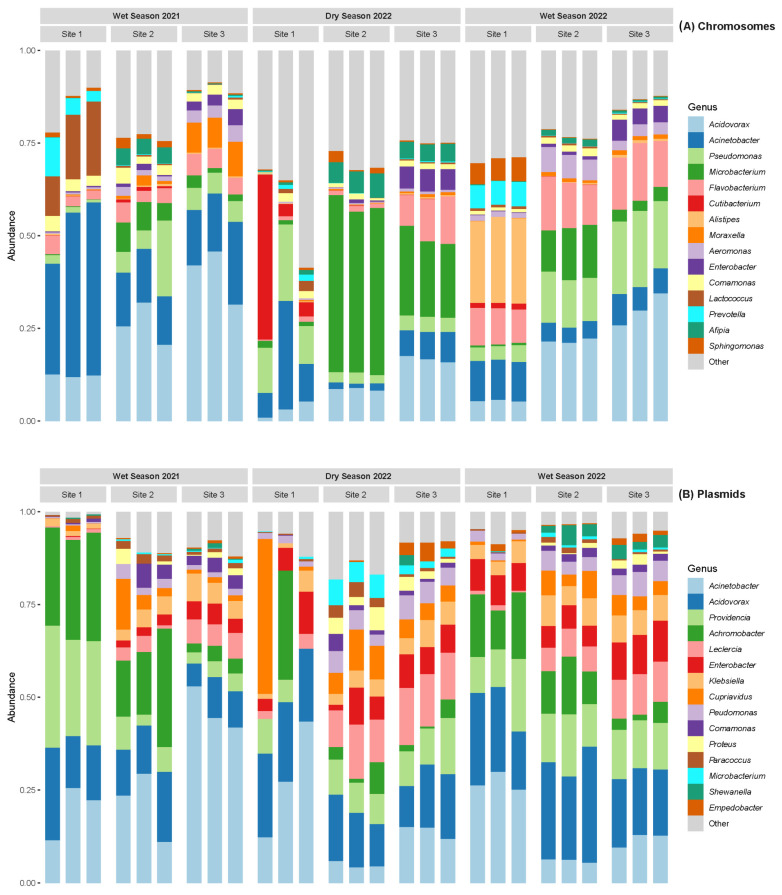
Relative abundance of the 20 most prevalent bacterial genera detected in chromosomal (**A**) and plasmid (**B**) fractions from sediment metagenomes collected during three sampling campaigns and sites along the Virilla River.

**Figure 3 antibiotics-14-00798-f003:**
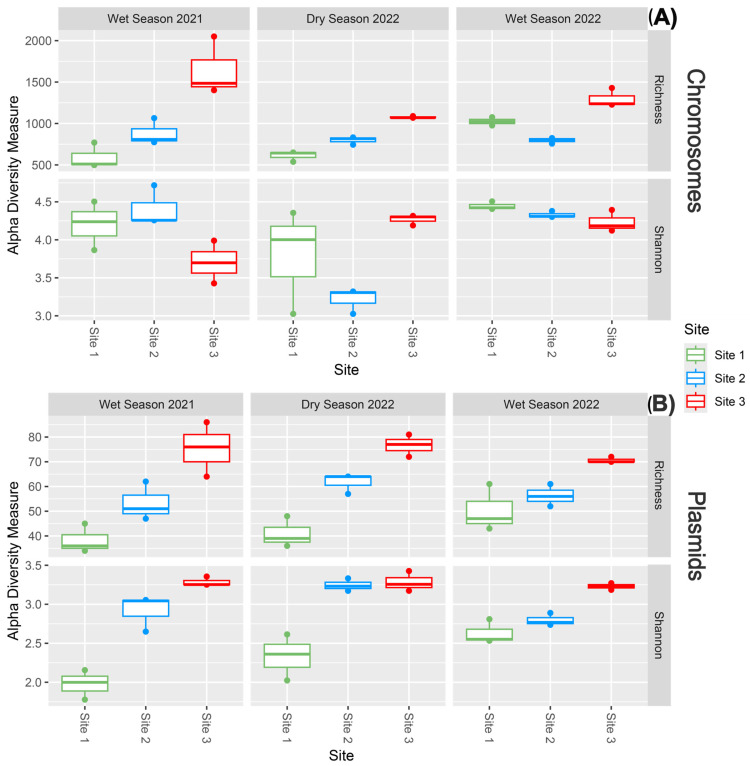
Alpha diversity plots show the number of taxa (richness) and the Shannon diversity index derived from (**A**) chromosomal and (**B**) plasmidic datasets of potential pathogenic genera of the sediment samples. Data is presented by sampling campaigns and sites.

**Figure 4 antibiotics-14-00798-f004:**
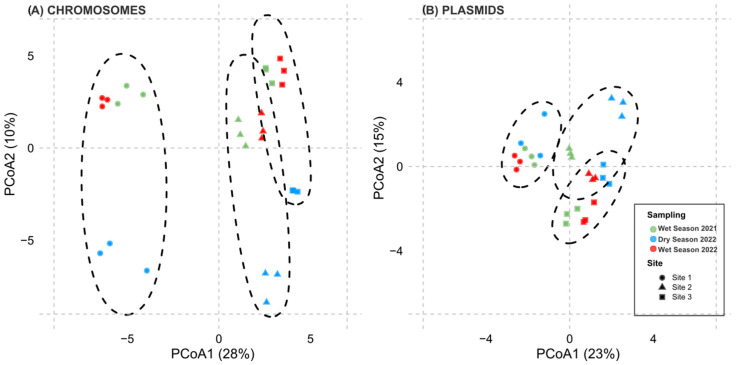
Principal coordinates analysis (PCoA) based on Bray–Curtis dissimilarity matrices computed from centered log-ratio (CLR) transformed taxonomic abundance profiles of potential pathogens in sediment samples. Panel (**A**) shows chromosomal profiles; Panel (**B**) shows plasmidic profiles. Colors indicate sampling campaigns (green = wet season 2021, blue = dry season 2022, red = wet season 2022), while shapes represent sampling sites. Ellipses denote sample variability within each site.

**Figure 5 antibiotics-14-00798-f005:**
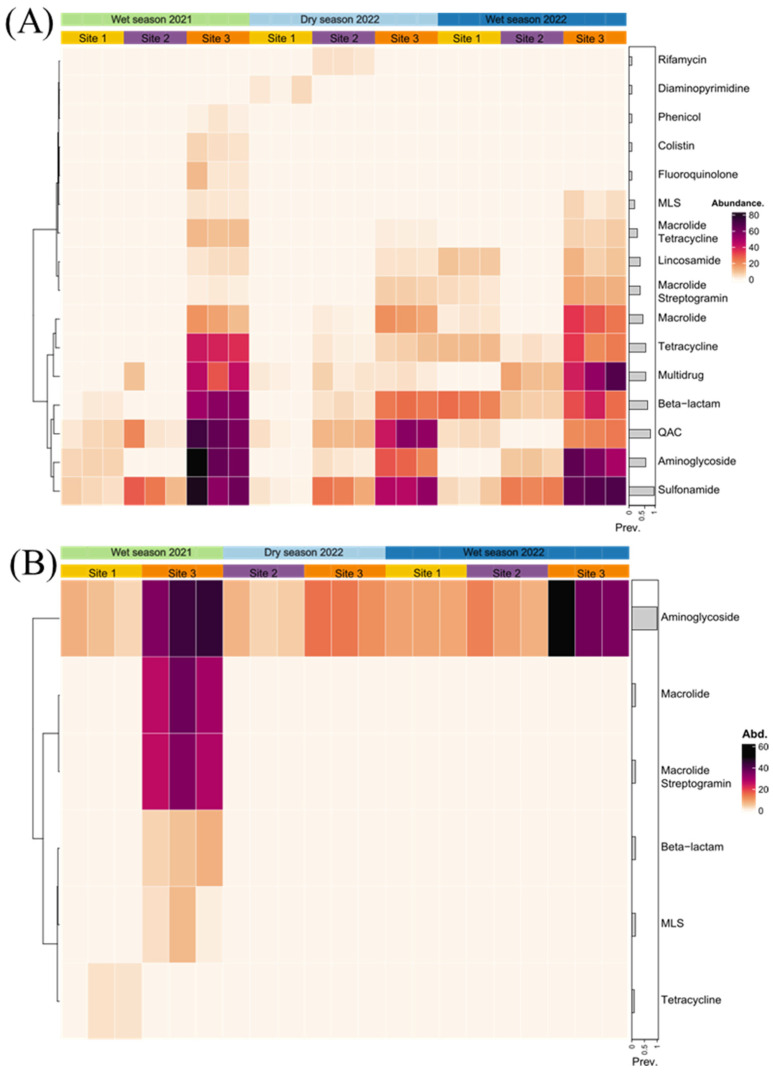
The abundance of antibiotic resistance genes (ARGs) in samples from the Virilla River. Panels (**A**) chromosomes and (**B**) plasmids present the abundance of ARGs in reads per kilobase million (RPKM) values. ARGs are grouped by categories based on their potential resistance phenotypes. Rows are reordered using a hierarchical clustering method, with bars on the right indicating the prevalence of ARGs across all samples. Only sampling sites where ARGs were detected are shown.

**Figure 6 antibiotics-14-00798-f006:**
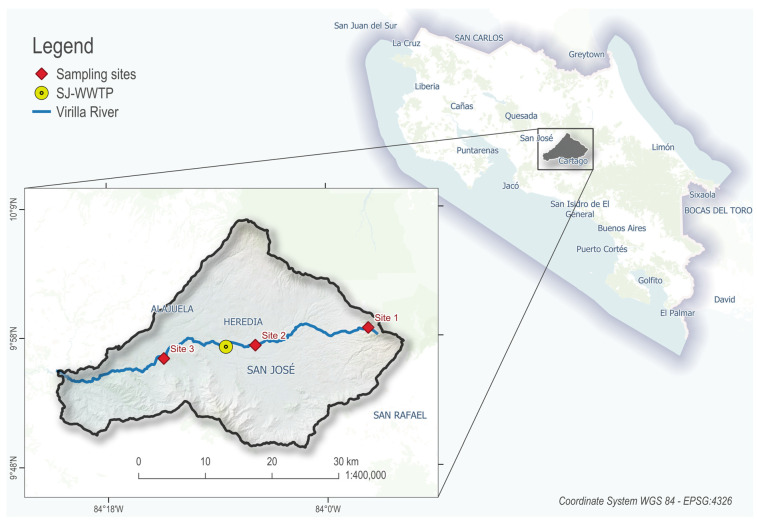
Virilla River watershed map. Locations of the sampling sites and the SJ-WWTP within the main course of the Virilla River.

**Table 1 antibiotics-14-00798-t001:** Physical–chemical and microbiological parameters of water samples in the Virilla River watershed.

Sampling Sites	Fecal Coliforms (Mean ± Standard Deviation) MPN/100 mL	*E. coli* (Mean ± Standard Deviation) MPN/100 mL	*E. faecalis* (Mean ± Standard Deviation) MPN/100 mL	O_2_ Sat % (Mean ± Standard Deviation)	Temperature (Mean ± Standard Deviation) °C	pH (Mean ± Standard Deviation)	Turbidity (Mean ± Standard Deviation) UNT	NSFQI
Sampling campaign 1 (wet season 2021)								
Site 1	(4.33 ± 7.76) × 10^4^	(2.99 ± 4.34) × 10^4^	(3.12 ± 1.90) × 10^4^	71.4 ± 3.7	18.7 ± 3.2	7.94 ± 2.30 × 10^−1^	142.0 ± 95.4	44
Site 2	(6.79 ± 10.50) × 10^4^	(6.79 ± 10.50) × 10^4^	(1.10 ± 1.65) × 10^5^	61.1 ± 33.0	19.5 ± 3.5	7.90 ± 2.23 × 10^−1^	33.0 ± 9.3	41
Site 3	(1.48 ± 1.49) × 10^5^	(1.35 ± 1.54) × 10^5^	(1.92 ± 12.30) × 10^4^	51.9 ± 6.4	19.2 ± 3.4	7.93 ± 9.50 × 10^−2^	13.1 ± 15.1	28
Sampling campaign 2 (dry season 2022)								
Site 1	(2.88 ± 0.64) × 10^2^	(2.11 ± 1.00) × 10^2^	(6.67 ± 3.29) × 10^2^	78.1 ± 0.0	15.8 ± 0.0	7.51 ± 1.09 × 10^−15^	9.8 ± 0.0	66
Site 2	(9.64 ± 7.22) × 10^4^	(6.71 ± 6.74) × 10^4^	(1.87 ± 0.44) × 10^4^	71.8 ± 0.0	23.4 ± 0.0	7.85 ± 1.09 × 10^−15^	8.7 ± 0.0	49
Site 3	(3.38 ± 1.10) × 10^5^	(2.59 ± 1.00) × 10^5^	(5.58 ± 2.90) × 10^3^	47.7 ± 0.0	28.0 ± 0.0	7.90 ± 1.09 × 10^−15^	21.9 ± 0.3	35
Sampling campaign 3 (wet season 2022)								
Site 1	(4.60 ± 0.00) × 10^3^	(4.60 ± 0.00) × 10^3^	(3.50 ± 0.00) × 10^3^	86.2 ± 0.0	15.9 ± 0.0	7.72 ± 0.0	38.0 ± 3.0	49
Site 2	(9.20 ± 0.00) × 10^4^	(5.40 ± 0.00) × 10^4^	(3.50 ± 0.00) × 10^4^	80.0 ± 0.0	20.7 ± 0.0	8.10 ± 0.0	167.0 ± 2.0	41
Site 3	(7.00 ± 0.00) × 10^4^	(3.10 ± 0.00) × 10^4^	(9.20 ± 0.00) × 10^3^	77.0 ± 0.0	25.6 ± 0.0	7.76 ± 1.09 × 10^−15^	215.0 ± 5.0	42

## Data Availability

The original contributions presented in the study are included in the article/Appendix A. Metagenomic data were uploaded to the National Center for Biotechnology Information (NCBI) repository (Bioproject accession no. PRJNA1240286).

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
