# Peer review of "Influence of Seasonality and Pollution on the Presence of Antibiotic Resistance Genes and Potentially Pathogenic Bacteria in a Tropical Urban River"

_antibiotics, 2025, doi:10.3390/antibiotics14080798_

Round 1
Reviewer 1 Report
Comments and Suggestions for Authors
Specific comments:
Line 573-574: ‘All water and sediment samples were kept cold during transport and analysed within 20 hours.’ Please specify how samples were kept cold (e.g., on ice-packs or in a portable fridge) and the approximate temperature.
Line 597: “Two hundred fifty milli-grams of the sample were used for sediments, and one filter (equivalent to 250 ml filtration 598 volume) was extracted for the water sample.” -This should be corrected to: '250 mg of the sample was used for sediments, and one filter (equivalent to 250 ml filtration volume) was extracted for the water sample.'
Line 614: Reference to Table S5 not found - Please ensure that Table S5 is provided or correctly cited.
Line 608-609: “The analysis targeted genes con-ferring resistance to multiple antibiotic classes, including sulfonamides, tetracyclines, beta-lactams, macrolides, quinolones, and trimethoprim.” – The sentence regarding antibiotic classes appears introductory and not methodological. It is recommended to include a detailed list of target genes and their corresponding primer sequences. If needed, provide this in a supplementary table.
Line 626: Typographical correction - 'to determine the insert size..' should be 'to determine the insert size.'
Line 647-648: 'A subset of data was generated...' - Please clarify which taxa were targeted as human pathogens. Listing specific genera or species would add clarity.
Line 662: Sentence incomplete - Please clarify or complete the phrase ‘and site of origin’.
Major Comments:
Introduction: Similar concepts are repeated throughout. The section lacks clearer structure and less redundancy. Consider including a descriptive comparison of previous work in the region to enhance relevance.
Methodology: The study area and sampling plan are not clearly explained. A descriptive table would help readers understand the design.
Conclusion: The conclusion currently lacks depth. Consider summarizing the main findings in relation to existing literature and highlighting broader implications for environmental AMR surveillance.
English: Overall, language is fluent while repetition of plural words breaks the interest of the reader. Careful editing can improve readability.
Overall summary:
This manuscript presents a comprehensive study of antibiotic resistance genes (ARGs) and potentially pathogenic bacteria along a tropical urban river in Costa Rica. By integrating qPCR and shotgun metagenomic sequencing across water and sediment matrices, and across seasonal and spatial gradients, the authors successfully demonstrate how plasmid and chromosomal ARGs behave differently under variable environmental pressures.
The study is well-designed, methodologically sound, and contributes valuable insights to the field of environmental microbiology and AMR surveillance. However, several areas require clarification, elaboration, and stylistic improvement to enhance readability and scientific impact.
Author Response
Please see the attachment. Also, both the revised manuscript and the Supplementary Material have been uploaded.

Reviewer 2 Report
Comments and Suggestions for Authors
Thank you for the opportunity to review the manuscript titled “Influence of Seasonality and Pollution on the Presence of Antibiotic Resistance Genes and Potentially Pathogenic Bacteria in a Tropical Urban River.” This is a timely and relevant study that adds valuable insights into seasonal and spatial dynamics of antibiotic resistance genes (ARGs) and microbial community composition in a tropical urban river ecosystem. The integration of qPCR and metagenomic approaches, with a distinction between chromosomal and plasmid-associated ARGs, enhances the strength and novelty of the work.
Major Comments:
- Lines 35–36: Kindly mention that sul-1, intl-1, and tetA are antibiotic resistance genes (ARGs) to avoid any confusion for readers from diverse backgrounds. Suggested revision:
“During the wet season 2021, the abundance of antibiotic resistance genes (ARGs) such as sul-1, intl-1, and tetA…” - It is advisable to avoid repeating words from the title in the keywords.
- The graphical abstract is well designed. However, I suggest shifting the legends below the figure, if feasible, to improve the visual appeal and ensure better readability.
- Why does the study include only three sampling campaigns, with two from the wet season (2021 and 2022) and only one from the dry season (2022)? Since the manuscript focuses on seasonal variation, a more balanced seasonal representation would provide stronger support for the conclusions. Kindly clarify if this design was due to logistical constraints or specific rainfall variations. Including this explanation in the Methods or Limitations section will help readers interpret the findings more clearly.
- While the manuscript discusses taxonomic diversity and mentions several potential pathogens a clear visual representation of overall taxonomic profiles is missing. It would greatly improve the manuscript if the authors could include a stacked bar plot or heatmap showing the dominant bacterial taxa (at phylum or genus level) in the chromosomal and plasmid fractions. Such a figure would complement the alpha/beta diversity results and help readers understand the broader community shifts across sites and seasons.
- The study identifies plasmid-associated ARGs, but it would be valuable to know whether multiple ARGs were co-localized on the same plasmid contigs, which could indicate potential multidrug resistance plasmids or hotspots for horizontal gene transfer. Were any co-occurrence patterns observed? If such an analysis were performed, please consider presenting it through a figure or table.
- The seasonal trends in ARGs are discussed well. However, linking these findings with broader ecological perspectives, such as the sink behaviour of sediments or the adaptive role of plasmids, could make the discussion stronger. Some points are repeated from the results; you may wish to streamline them.
- The conclusion is appropriate. However, it will be good to add one or two lines on how this study can support policy decisions or public health monitoring.
English editing is recommended before final acceptance.
Author Response
Please see the attachment. Also both the revised manuscript and the Supplementary Material have been uploaded.

Reviewer 3 Report
Comments and Suggestions for Authors
The study provides valuable insights into ARG dynamics in tropical rivers but requires substantial revision as listed below.
- DNA extraction was done using the same kit (DNeasy PowerSoil Pro) for water and sediment samples but with different starting masses (250 mg sediment vs. filter from 250 mL water). Using only 250 mL of water may not be enough for further downstream processing.
- BF >1 supports the alternativehypothesis (not null, as stated in Methods 4.9).
- Physicochemical parameters (e.g., temperature, DO) vary significantly by site/season (Table 1) but weren’t included as covariates in ARG models.
- The conclusion that plasmids rapidly respond to environmental stress (p. 15) lacks mechanistic evidence (e.g., conjugation rates or plasmid copy number). Observational metagenomic data alone cannot support causality.
- Sampling occurred during La Niña (p. 17), which increased rainfall in wet season 2022 vs. 2021. This climatic anomaly isn’t incorporated into seasonal comparisons.
- Relative abundance scales (0.00001–10) lack units (assumed copies/16S rRNA). Clarify if normalized to 16S rRNA in caption.
- "NSFQI" values are reported, but the index calculation (parameters/weights) isn’t described. Standard NSF-WQI uses 9 parameters; here, only 5 are listed.
- Table S3 (ARG list) is critical but not provided. Accessibility must be confirmed.
- "Plasmidome" (p. 4) is used interchangeably with "plasmid-associated" genes. Define clearly to avoid confusion.
- "Fair" water quality (p. 6) for Site 1 in dry season 2022 conflicts with NSFQI=66 ("fair" is 50.1–70), but Site 3=35 ("poor"). Emphasize that all sites fall below "good" quality.
Author Response

(The authors gave the same response as above.)

Reviewer 4 Report
Comments and Suggestions for Authors
The study addresses antibiotic resistance in tropical urban aquatic systems—an underrepresented but highly relevant field in the global AMR discourse.
Major Issues:
- The results based on the figure seems that the quantity of ARGs is increasing with year. The paper just compares with wet and dry season, but it can also be associated with year. Given that dry season span just over 1 year, this should be included as potential cofounder.
- More contextual metadata would strengthen interpretations, like intensity of rainfall, hospitals and agricultural sources nearby which may be causing the increase in AMR.
- "In the chromosomic dataset, we identified 11,477 taxa representing 1,966 bacterial 223 genera, of which 251 genera (4,044 taxa or 35%) were classified as potential human 224 pathogens." - The current pathogens defined is less than 4000. [Ref: https://www.nature.com/articles/nrmicro2644] The authors need to clarify how this was done, maybe use preexisting sources like The Microbe Directory [https://www.biorxiv.org/content/10.1101/2019.12.20.860569v1.abstract]
- For better understanding, a control would make quantification and understanding of this paper better - for example what are the ARGs we usually see in environment.
Author Response

(The authors gave the same response as above.)

Round 2
Reviewer 1 Report
Comments and Suggestions for Authors
Dear Authors,
Thank you for your careful revision. The manuscript has improved in clarity and structure, and the study presents interesting findings on the seasonal dynamics of antibiotic resistance genes in a tropical urban river system. However, a few issues remain that should be addressed:
Line 580 - Please change the word “totaling” to the British English spelling “totalling” for consistency with the rest of the manuscript.
Table 2 placement - Consider moving Table 2 to the Supplementary Material, as it primarily contains site metadata and would be more appropriately placed there. A reference to it in the main text would still provide clarity.
Capitalization consistency - There is inconsistency in the use of uppercase letters, especially in section and subsection titles (e.g., “Antibiotic Resistance Genes Detection from sediment samples”). Please ensure uniform capitalization, either sentence case or title case, across all headings and figure/table titles.
Writing style - The manuscript is currently written in a somewhat explanatory and narrative tone. Scientific writing should be more concise and descriptively precise. The Methods, Results, and Discussion sections in particular would benefit from clearer structure, reduced redundancy, and tighter phrasing to maintain scientific focus.
Language and readability - While the manuscript conveys important information, the writing style in several parts reduces reader engagement. I strongly recommend a thorough language edit by a native English speaker or a professional scientific editing service to enhance readability and overall presentation.
Best
Comments on the Quality of English LanguageThe manuscript requires substantial language improvement. Several sections are overly explanatory and lack precision, which detracts from the scientific clarity. A professional English-language edit is strongly advised to address grammar, structure, and formatting issues before publication.
Author Response
For research article: Influence of Seasonality and Pollution on the Presence of Antibiotic Resistance Genes and Potentially Pathogenic Bacteria in a Tropical Urban River
Response to Reviewer 1 Comments Round 2
|
||
1. Summary |
|
|
Thank you very much for taking the time to review this manuscript. Please find the detailed responses below and the corresponding revisions/corrections highlighted in yellow in the re-submitted files.
|
||
2. Questions for General Evaluation |
Reviewer’s Evaluation |
Response and Revisions |
Does the introduction provide sufficient background and include all relevant references? |
Can be improved |
The reviewer’s comments have been incorporated into the manuscript and are detailed below in this document, along with a specific response to each comment. |
Are all figures and tables clear and well-presented?? |
Yes |
Every modification has been marked in yellow in the main text |
Is the research design appropriate? |
Can be improved |
|
Are the methods adequately described? |
Can be improved |
|
Are the results clearly presented? |
Can be improved
|
|
3. Point-by-point response to Comments and Suggestions for Authors
|
||
Specific Comment 1: Line 580 - Please change the word “totaling” to the British English spelling “totalling” for consistency with the rest of the manuscript. |
||
Response 1: Thank you for your observation. We have corrected the spelling by changing “totaling” to the British English spelling “totalling,” following the style used throughout the manuscript. This modification can be seen on page 18, line 581.
|
||
Specific Comment 2: Table 2 placement - Consider moving Table 2 to the Supplementary Material, as it primarily contains site metadata and would be more appropriately placed there. A reference to it in the main text would still provide clarity.
|
||
Response 2: Thank you for the suggestion. Table 2, which contains site metadata, has been moved to the Supplementary Material and is now presented as Table S4. A reference to this supplementary table has been added in the main text to maintain clarity for the reader. This modification can be seen on page 18, lines 584-585.
Specific Comment 3: Capitalization consistency - There is inconsistency in the use of uppercase letters, especially in section and subsection titles (e.g., “Antibiotic Resistance Genes Detection from sediment samples”). Please ensure uniform capitalization, either sentence case or title case, across all headings and figure/table titles.
Response 3: Thank you for your valuable observation regarding capitalization consistency. We have carefully reviewed the manuscript and applied the requested corrections to ensure uniform capitalization. Specifically, we have standardized the use of sentence case across all sections and subsection titles, as well as in the titles and captions of all tables and figures.
Specific Comment 4: Writing style - The manuscript is currently written in a somewhat explanatory and narrative tone. Scientific writing should be more concise and descriptively precise. The Methods, Results, and Discussion sections in particular would benefit from clearer structure, reduced redundancy, and tighter phrasing to maintain scientific focus. Response 4: We appreciate the reviewer’s comment regarding the writing style of the manuscript. In response, we conducted a comprehensive revision of the text to improve clarity, conciseness, and scientific precision, particularly in the Methods, Results, and Discussion sections. We have carefully edited the manuscript to reduce narrative tone, eliminate redundancy, and strengthen descriptive accuracy while maintaining the scientific content and the changes previously approved by the other reviewers. Given the scope of this language revision, the specific pages and line numbers for all changes are not listed here. However, all modifications made in response to this comment have been highlighted in yellow throughout the revised document for ease of review.
Specific Comment 5: Language and readability - While the manuscript conveys important information, the writing style in several parts reduces reader engagement. I strongly recommend a thorough language edit by a native English speaker or a professional scientific editing service to enhance readability and overall presentation.
Response 5: We thank the reviewer for this valuable observation regarding language and readability. In response to the suggestion, we conducted a thorough language review of the entire manuscript using an advanced AI-based scientific editing tool to enhance clarity, improve flow, and ensure consistency throughout the text. Additionally, most of the authors are fluent in English and have published several scientific articles in peer-reviewed international journals. Nevertheless, we took this opportunity to refine the manuscript further and elevate the quality of its presentation. Given that the language revision was applied across the entire manuscript, we do not provide specific page or line numbers for each modification in this response. However, all changes have been highlighted in yellow in the revised version to facilitate the review process. We have also ensured that all previously approved modifications by the other reviewers were respected.
|
||
The authors confirm that a thorough revision of the English language was conducted throughout the manuscript to ensure clarity, accuracy, and consistency using Generative IA. The authors also verify and take full responsibility for the use of generative artificial intelligence in the preparation of this manuscript. Generative AI was used exclusively to improve the English writing, as English is not the authors’ native language. AI was not used to generate scientific content, ideas, or to analyze or interpret any data.
|
Reviewer 2 Report
Comments and Suggestions for Authors
The authors have satisfactorily addressed all my queries, and the suggested revisions have been appropriately incorporated; the manuscript can now be considered for publication.
Author Response
Thank you very much for your valuable comments, which were important for improving the quality of the article.
Reviewer 3 Report
Comments and Suggestions for Authors
I am satisfied with the revised manuscript
Author Response

(The authors gave the same response as above.)
